# Crystallographic and Computational Electron Density of d_x2-y2_ Orbitals of Azo-Schiff Base Metal Complexes Using Conventional Programs

**DOI:** 10.3390/molecules26030551

**Published:** 2021-01-21

**Authors:** Yuji Takiguchi, Yuika Onami, Tomoyuki Haraguchi, Takashiro Akitsu

**Affiliations:** Department of Chemistry, Faculty of Science, Tokyo University of Science, 1-3 Kagurazaka, Shinjuku-ku, Tokyo 162-8601, Japan; 2317064@ed.tus.ac.jp (Y.T.); 1319526@ed.tus.ac.jp (Y.O.); haraguchi@rs.tus.ac.jp (T.H.)

**Keywords:** Schiff base, metal complex, electron density, coordination bond, DFT

## Abstract

The crystal structures of two azobenzene derivative Schiff base metal complexes (new C_44_H_40_CuN_6_O_2_ of *P*-1 and known C_44_H_38_MnN_6_O_7_ of *P*2_1_/c abbreviated as Cu and Mn, respectively) were (re-)determined experimentally using conventional X-ray analysis to obtain electron density using a PLATON program. Cu affords a four-coordinated square planar geometry, while Mn affords a hexa-coordinated distorted octahedral geometry whose apical sites are occupied by an acetate ion and water ligands, which are associated with hydrogen bonds. The π-π or CH-π and hydrogen bonding intermolecular interactions were found in both crystals, which were also analyzed using a Hirshfeld surface analysis program. To compare these results with experimental results, a density functional theory (DFT) calculation was also carried out based on the crystal structures to obtain calculated electron density using a conventional Gaussian program. These results revealed that the axial Mn-O coordination bonds of Mn were relatively weaker than the in-plane M-N or M-O coordination bonds.

## 1. Introduction

X-ray crystal structure analysis is powerful as a means of determination of structures as well as a means of measurement of charge density for chemical substances in the solid state. From the obtained electron density, the state of the covalent bonds and coordinate bonds and the characteristics of intramolecular/intermolecular interactions can be clearly estimated. In recent years, due to the improvement of experimental accuracy and progress of basic theory, electron densities using the X-ray data have been precisely discussed for both metal complexes and metalloproteins. High-resolution structural analysis has been employed for valence electrons and charge transfer around metal sites in these proteins [1,2]. In X-ray crystal structure analysis, in principle, atoms are located and assigned in the places indicating high electron densities from the Fourier transferred data obtained by experiments to build and refine structural models [3], which results in empirical evaluation of bond lengths and intermolecular interactions [4]. Additionally, with the development of Hirshfeld surface analysis, it is possible to know the breakdown of intermolecular interactions in a crystal packing [5] especially for crystalline organic compounds [6] using freely available programs.

On the other hand, multipole expansion method [7] and X-ray atomic (or molecular) orbital analysis [8] have been used to experimentally examine the shapes of valence electrons and orbitals from electron density. With these methods, we can see that the approximate distribution of the total electron density is shaped like an atomic orbital (like a spherical harmonic). For example, studies have been reported using these methods to discuss the d-orbitals of hexacoordinate octahedral complexes [9]. Meanwhile, quantum crystallography, which uses quantum mechanics and computational chemistry to refine X-ray crystal structure analysis, is also attracting attention [10]. By representing wave functions from electron densities obtained from X-ray crystal structure analysis, refinement of crystal structures using computational chemistry such as density functional theory (DFT) has been carried out [11,12] using advanced and specially dedicated programs.

Information on bonds obtained from X-ray crystallography is mainly determined from the interatomic distance. Determining the bond from the information obtained from the electron density requires specialized knowledge and skill. In the first place, chemical bonds are expressed by the overlap of wave functions and are defined quantum mechanically. Discussing bonds in quantum chemistry calculations seems to be easier than discussing bonds crystallographically. Therefore, in this study, we performed quantum chemistry calculations based on the molecular structure obtained by the experiment, and tried to compare the experiment with the calculation and supplement the experiment through calculations.

In this context, we tried to employ “conventional programs” for DFT calculations and Hirshfeld surface analysis for comparison with electron density, including intermolecular interactions based on experimental X-ray crystal analysis, which was visualized with a commonly distributed program in this field. The test samples were two Schiff base metal complexes having an azobenzene moiety (new **Cu** of *trans*-[CuN_2_O_2_] and known Mn of *cis*-[CuN_2_O_2_X_2_]). We focused on their photochemical behavior [13,14,15] to make the discussion of bonding features as simple as possible. 

## 2. Materials and Methods

### 2.1. X-ray Crystallography

Green prismatic single crystals (0.60 × 0.30 × 0.20 mm^3^) of new Cu were obtained in a similar way to that in the literature [16] and analyzed, while the present analysis of Mn was a re-determination of CCDC 1457179 [17] to obtain electron density data (Appendix A). The samples were set on top of a glass capillary, coated with a thin layer of Araldite epoxy resin. Intensity data were collected on a Bruker APEX2 CCD diffractometer (Bruker, Billerica, MA, USA) with Mo-Kα radiation monochromated by graphite (λ = 0.71073 Å). Data treatment used the program package SAINT (Bruker, Billerica, MA, USA). An empirical absorption correction for intensity was applied by the program SADABS (Bruker, Billerica, MA, USA). In this program package, the structures (phase problem) were initially solved by direct methods with a SHELXS-97 [18], expanded by Fourier techniques, and finally refined by full-matrix least-squares methods based on *F*^2^ using a SHELXL-97 program [18]. All non-hydrogen (heavy) atoms were readily located to construct a model and were refined by anisotropic (thermal) displacement parameters. Hydrogen atoms were located at geometrically calculated positions and refined using riding models, except for the -COOH groups for Mn. 

Crystallographic data for **Cu** (CCDC 2051658): C_44_H_40_CuN_6_O_2_, Triclinic, space group *P-1* (#2), *Z* = 4, *a* = 8.5620(10) Å, *b* = 21.508(3) Å, *c* = 21.526(3) Å, *α* = 87.817(2)°, *β* = 78.555(2)°, *γ* =78.710(2)°, *V* = 3810.0(8) Å ^3^, *ρ*_calc_ = 1.190 gcm^−3^, *μ* = 0.619 mm^−1^, *F*(000) = 1748, *S* = 0.514, *R*_1_[*I*>2σ(*I*)] = 0.0494, *wR2* = 0.0893, *T* = 173 K.

Crystallographic data for Mn: C_44_H_38_MnN_6_O_7_, Monoclinic, space group *P*2_1_ (#4), *Z* = 4, *a* = 12.075(8) Å, *b* = 16.699(12) Å, *c* = 23.172(16) Å, *β* = 94.540(5)°, *V* = 4658.0(6) Å ^3^, *ρ*_calc_ = 1.166 gcm^−3^, *μ* = 0.336 mm^−1^, *F*(000) = 1700, *S* = 1.115, *R*_1_[*I*>2σ(*I*)] = 0.0898, *wR2* = 0.2711, *T* = 173 K.

### 2.2. Calculations

The calculations of the crystal structures of Cu and Mn were carried out using the Gaussian 09W software package Revision D.02 (Gaussian, Inc., Wallingford, CT, USA) [19] with a Windows 10 personal computer. Since the salen-type ligand is considered to have a weak crystal field, the calculation was performed assuming a d^9^ for Cu(II) and high spin d^4^ for Mn(III). The calculation strategy was based on previous studies of Schiff base complexes [20,21]. Density functional theory (DFT) was used together with B3LYP functional for all calculations because of the balance between calculation accuracy and calculation time. The basis set Lanl2DZ (Cu, Mn) was applied to the central metal (Cu, Zn), and the basis set 6-31G (d) was applied to the other atoms (C, O, N, H). ECP (effective core potential) was applied to the central metal. GaussView5 was used for analysis and visualization of the calculation results. All calculations were done under gas phase (isolated) conditions. Single-point calculation was performed on the structure obtained by X-ray crystal structure analysis, and Mulliken charge and spin density were calculated. Natural bond orbital (NBO) analysis [22] was performed to estimate bond orders. 

The CRYSTAL EXPLORER [23] program was used for Hirshfeld surface analyzes and fingerprint plots [24,25]. The Hirshfeld surface was represented by the normalized contact distance (*d*_norm_). If it is shorter than the van der Waals radius, it is shown in red, and if it is longer, it is shown in blue. In two-dimensional (2D) fingerprint plots, *d*_e_ was plotted on the vertical axis and *d*_i_ was plotted on the horizontal axis.

## 3. Results and Discussion

### 3.1. Brief Description of Crystal Structures

Mn affords a hexa-coordinated distorted octahedral geometry, with in-plane oxygen and nitrogen coordination atoms, and axial water molecules and acetate ions (Figure 1). In the equatorial position, the two oxygen and nitrogen atoms are coordinated on the same side, forming a *cis*-form. The in-plane Mn-N bond length is longer than that of Mn-O. Since O2, O3, N3, and N4 are on the square centered on the Mn atom, the three-dimensional distortion of C22 and C15 is large. The O5-Mn-O7 angle of the two oxygen atoms in the axial position is 170.5 (2)° deviated from linear 180°. Moreover, it is considered that the two oxygen atoms depend on the N4 and N3 sides and avoid electrostatic repulsion with O2 and O3. The N-N-C-C dihedral angle of the azo group is 173°, which is more than the ideally linear angle of 180°. In the packing, the two Mn complexes are parallel at an angle of about 73° (Figure 2a). The ideal angle of the benzene π-π interaction is known to be 63° [26], which is close to that seen with the two Mn complexes. The Mn complex during packing is considered to be stabilized by π-π interactions. In addition, the two Mn molecules form dimers with a water molecule at the axial position between the two -OCH_3_ groups of the other complex (Figure 2b). 

Hydrogen bonds between water molecules and three -OCH_3_ groups are also found. In addition, since the adjacent -OCH_3_ group is sandwiched between the two phenyl groups on the N3 and N4 sides, it is considered that there is also an CH-π interaction [26] (Figure 2c). The upper and lower azo groups have different dihedral angles, and the distance between hydrogen and the benzene ring is close, which suggests a weak interaction (Figure 2d). 

Cu affords a four-coordinated square planar geometry, in which O and N are lined up diagonally to be a trans-form. The two ligands are coordinated oppositely (Figure 3). Comparing Cu-O and Cu-N, Cu-N has a longer bond length. The dihedral angle of the azobenzene of Cu is 178.57°, which is closer to 180° than that of Mn, and it can be seen that it is stabilized by the extension of the π-conjugated system. The phenyl(C)-C-C-N dihedral angle is 64.01° and is a gauche type arrangement. At first glance, the conformation has a large steric hindrance, but it is also possible that the phenyl group is folded to give an advantage in packing and due to the influence of the CH-π interaction between molecules. Considering the packing, it can be seen that there is a π-π interaction because the two complexes are lined up in parallel. The angle is 65.6°, which is close to the ideal angle. The phenyl group extending from N3 is considered to have CH-π interactions with the upper and lower complexes (Figure 4).

As shown in Figure 5 and Figure 6, the electron density diagram by crystal structure reveals that there is a region with high electron density in the center, which is the central metal. In both complexes, the electron density is cyclically distributed and the π-conjugated system is widespread. The N and O are indistinguishable from the contour map because their atomic numbers are close. The higher value of O is ascribed to a higher electronegativity than that of C. 

### 3.2. Computational Results

From single point calculations, the total energy of Cu was calculated as −2375.27 a.u., and the total energy of Mn was calculated as −2657.34 a.u. The HOMO-LUMO levels of both complexes are tabulated in Table 1. Mulliken charge and spin density around the central metal are listed in Table 2.

As shown in the electron density and molecular orbital (MO) diagram from the DFT calculations (Figure 5b, Figure 6b and Figure 7) based on the calculated Mulliken charge, there is a region with high electron density centered on the central metal. It can be confirmed that the electron density is delocalized in the aromatic ring and the charge is spread throughout. The difference in electronegativity of C, N, and O can be recognized from the figure. The spin density of the central metal was calculated to be 0.558 for Cu and 3.75 for Mn. The electron densities from crystal structure analysis are in good agreement with those from computation. In both cases, the electron density of the central metal is high. The π-conjugated system of Schiff bases is also extended in both Cu and Mn. It should be noted that the calculated electron density reproduced the experimental results well. The difference between the electron densities of O and N is hardly visible on X-rays, but the difference is clearly visible on the contour map by DFT, indicating that the charge density of O is high. Distorted contour lines are observed in Figure 5a and Figure 6a, on the other hand, Figure 5b and Figure 6b shows contour lines with clear peaks and (low-value) valleys. In general, crystal structure analysis is performed at finite temperature and under normal pressure with vibration due to thermal fluctuations of atoms and valence electrons, providing time-averaging electron density. On the other hand, in general, DFT calculation is carried out at 0 K and in a vacuum. This would be the difference between Figure 5a,b and Figure 6a,b.

The most important approximation in this study was to calculate the molecules in the crystal, which is a condensed system, as the molecules in the gas phase, which is an isolated system. In DFT calculation, it is common to calculate the energy, etc. after finding the optimized structure of the molecule. These are the optimized structures in the gas phase. Calculations using solvation approximation are also implemented in programs such as Gaussian, but it is still difficult to calculate the molecular structure in a solid from the first principle. Therefore, in this calculation, the structure in the crystal obtained by crystal analysis is considered as the optimized structure in the crystal, and this is calculated in the gas phase (isolated system). There is an intermolecular interaction in the crystal, and van der Waals force and hydrogen bonds are generated. These interactions are expected to change the electronic state of the molecule, but the central metal surrounded by ligands is less affected, indicating that these are good approximations.

Table 3 shows the Wiberg bond index obtained by NBO analysis. In Cu, the binding index of Cu-O is larger than that of Cu-N, indicating that the binding is strong. It can also be confirmed that the binding force is equal on the left and right. In Mn, the bond index between the O atom of the water molecule and Cu is small, so it can be seen that there is almost no bond or the bond is weak. The O atom of the acetic acid molecule is considered to be bonded because the bond index is similar to that of other atoms. Comparing O and N atoms, it can be confirmed that the O atom is strongly bonded. From the above, it is suggested that Mn is a five-coordinated complex in which Schiff bases are coordinated to the left and right and acetate ions are coordinated from above.

The calculated Hirshfeld surface is depicted in Figure 8. Points contributing to the surface were scaled to blue, then green, then red for increasing contribution. Two red areas (shorter than the van der Waals radius) can be seen in Cu. The mark (1) is considered to be the interaction between π-conjugated systems, and (2) is considered to be the CH-π interaction between aromatic rings and H atoms. In Mn, there is a red area near (3). It is considered that this is due to the hydrogen bond between the water molecule and the H atoms. Mark (4) is considered to be the interaction between the aromatic ring and the terminal H of the azo-group.

In fingerprint plots for Cu and Mn (Figure 9 and Figure 10, respectively), the H ... H interaction is 50% or more, and it can be seen that the van der Waals force is dominant. A characteristic wing can be seen in the fingerprint plot of **Cu**, confirming the presence of a CH-π interaction. A dark part (green) is seen near *d*_e_ = *d*_e_ = 1.8 Å, which can be said to be due to the π-π interaction. Since the central metal is four-coordinated, the equatorial position is also spatially vacant. Therefore, it is considered that there are Cu∙∙∙N interactions and Cu∙∙∙H interactions. Wings can be seen in Mn as well, confirming the existence of a CH-π interaction. A π-π interaction can be confirmed around 1.8 Å. Unlike Cu, since Mn is six-coordinated, it is sterically bulky and the ratio of π-π interactions is small. The ratio of O∙∙∙H interactions is as high as 14%, showing sharp characteristics due to hydrogen bonds. It is thought that this is due to the hydrogen bond between the hydrogen of the water molecule and that of the -OCH_3_ group. For both Cu and Mn, the reason may be van der Waals force.

In **Cu**, π-π interactions and CH-π interactions, like polycyclic aromatics, were also confirmed due to the structure of the conjugated system with an extended azo-salen-type ligand. Since the phenyl group is substituted on the Schiff base, an interaction between the upper and lower parts, which is not seen in polycyclic aromatics, was also observed, which is considered to affect the packing. In addition, since the transition metal is contained in the center, interactions such as metal → H and metal ← N are considered, and the result is expected to be a new possibility of packing polycyclic aromatics. At first glance, it looks like there is a hexa-coordinated complex in Mn, but it was discovered that a dimer was formed by the π-π interaction and water molecules were taken into the vacant space composed of Mn, the O atom in the skeleton of salen-type ligands, and the -OCH_3_ group. Schiff bases with -OCH_3_ groups are used as ligands for 3d-5f trinuclear complexes [27,28]. In this experiment, the water molecules entered the space where the lanthanoid was coordinated and crystallized. If this happens in a small molecule, the water may enter the space, and various crystallographic possibilities such as changes in structure, packing, and physical properties due to differences in molecules are expected.

## 4. Conclusions

In this study combining crystallography and quantum chemistry calculations, the electron density and molecular orbital of a Schiff base complex having an azobenzene derivative ligand were compared and explained. In the electron density distribution map from the crystal structures, the shape of the orbital of the central metal cannot be seen, and a specialized program is required, but valence electrons and d-orbitals can be easily displayed by performing quantum chemistry calculations. NBO analysis showed that the binding of water molecules was quite weak in Mn. Computational chemistry was able to provide the information that could not be obtained by crystal structure analysis. However, in this calculation, the molecule is placed in an isolated system, and it is considered that the molecule placed in the crystal cannot be completely reproduced. In order to have a more accurate discussion, it is necessary to develop calculation conditions that reproduce the inside of the crystal and to establish a calculation method that incorporates periodic boundary conditions. In this way, discussion of coordination and hydrogen bonds based on data of X-ray crystal structure analysis is possible to some extent by using conventional programs.

## Figures and Tables

**Figure 1 molecules-26-00551-f001:**
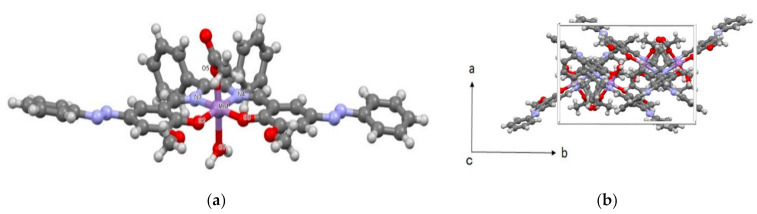
Molecular (**a**) and crystal (**b**) structures of Mn.

**Figure 2 molecules-26-00551-f002:**
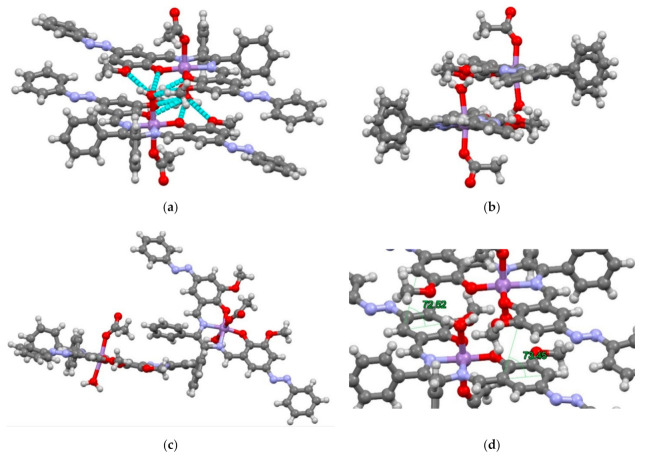
Crystal structure of Mn. (**a**) hydrogen bonds of water molecules, (**b**) dimers of complexes, (**c**) CH-π interactions, and (**d**) π-π interactions.

**Figure 3 molecules-26-00551-f003:**
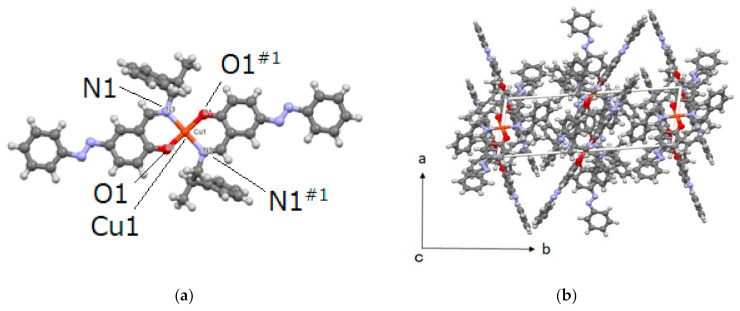
Molecular (**a**) and crystal (**b**) structures of Cu. Selected bond distances [Å] and angles [°]: Cu1-O1 = 1.890(3), Cu1-N3 = 2.008(3). O1-Cu1-N3 = 91.65(13), and O1-Cu1-N3^#1^ = 88.35(13) (Symmetry code: #1 (−x + 2, −y, −z + 1)).

**Figure 4 molecules-26-00551-f004:**
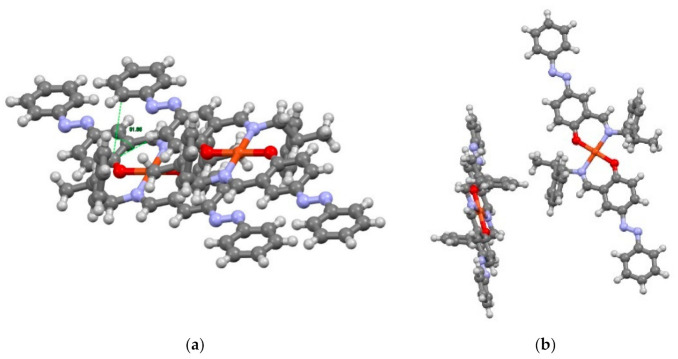
π-π interaction (**a**) and CH-π interaction (**b**) of Cu.

**Figure 5 molecules-26-00551-f005:**
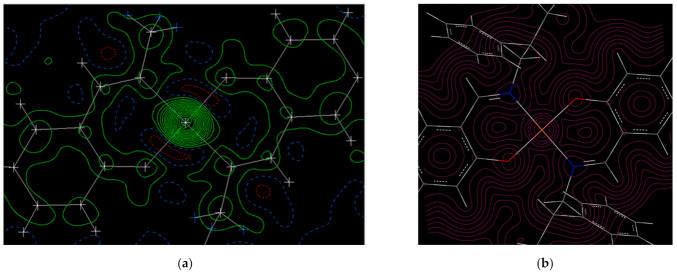
Fo maps (**a**) and electron density diagrams from the density functional theory (DFT) (**b**) for Cu.

**Figure 6 molecules-26-00551-f006:**
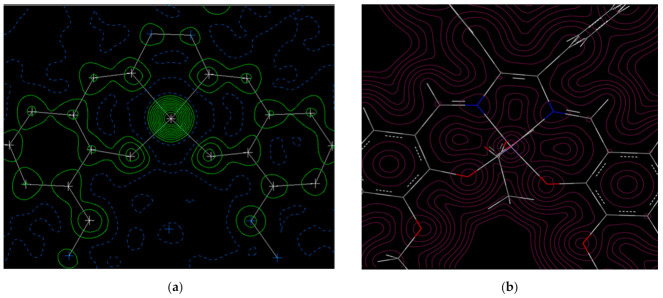
Fo maps (**a**) and electron density diagrams from the DFT (**b**) for Mn.

**Figure 7 molecules-26-00551-f007:**
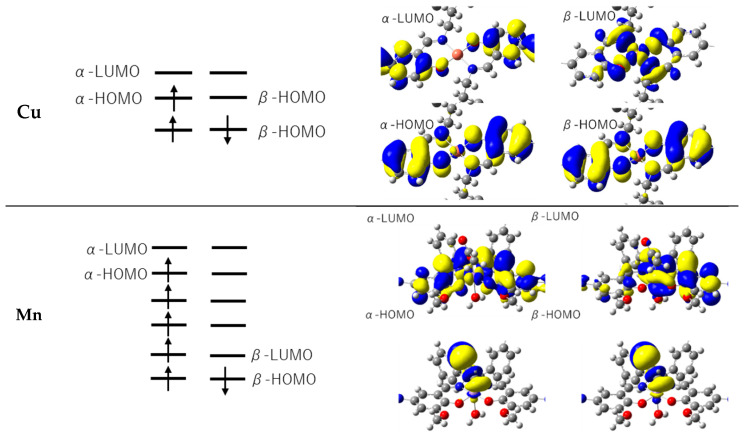
Molecular orbital diagrams of Cu and Mn.

**Figure 8 molecules-26-00551-f008:**
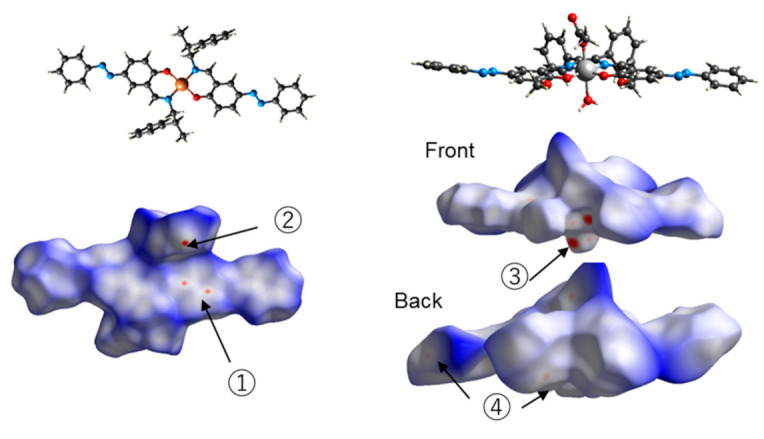
Hirshfeld surface of Cu (**right**) and Mn (**left**) mapped by *d*_norm_.

**Figure 9 molecules-26-00551-f009:**
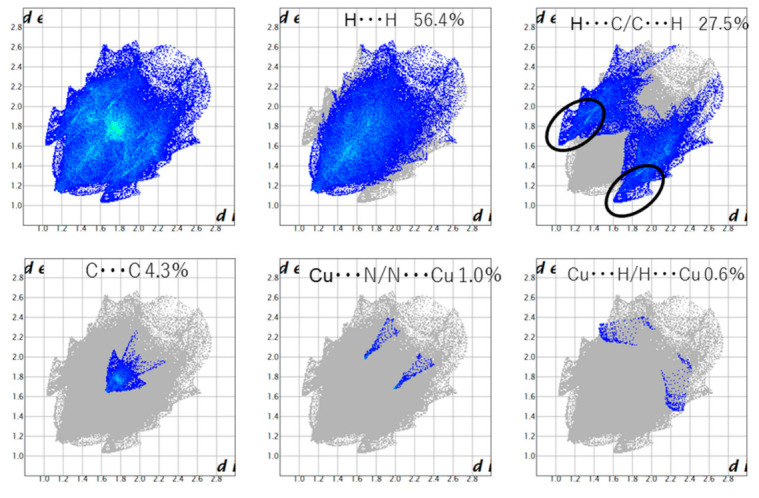
Fingerprint plots for **Cu**.

**Figure 10 molecules-26-00551-f010:**
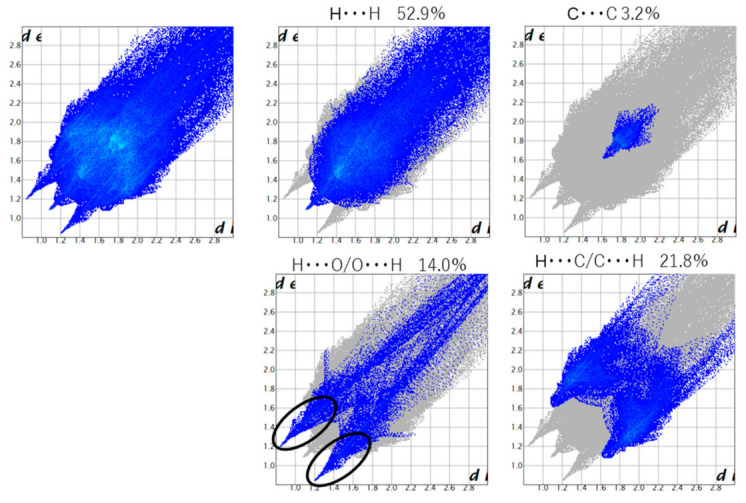
Fingerprint plots for Mn.

**Table 1 molecules-26-00551-t001:** HOMO-LUMO levels (a.u.).

	α-HOMO	β-HOMO	α-LUMO	β-LUMO	
Cu	−0.19582	−0.19432	−0.06215	−0.08883	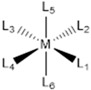
Mn	−0.14569	−0.08506	−0.14512	−0.08739

**Table 2 molecules-26-00551-t002:** Mulliken charge (1) and spin density (2) around the central metal.

		M	L1	L2	L3	L4	L5	L6
Cu	(1)	0.645	−0.469	−0.635	−0.469	−0.635	-	-
	(2)	0.558	0.102	0.110	0.102	0.110	-	-
Mn	(1)	0.932	−0.609	−0.560	−0.523	−0.629	−0.491	−0.767
	(2)	3.75	0.0307	−0.0357	−0.0327	0.0352	0.121	0.0161

**Table 3 molecules-26-00551-t003:** Wiberg bond index between M and Ln.

	L1	L2	L3	L4	L5	L6
Cu	0.1158	0.1340	0.1159	0.1340	-	-
Mn	0.1830	0.1239	0.1217	0.1943	0.1762	0.0616

## Data Availability

The data presented in this study are openly available in the article and Appendix A.

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
