# Peer review of "Crystallographic and Computational Electron Density of dx2-y2 Orbitals of Azo-Schiff Base Metal Complexes Using Conventional Programs"

_molecules, 2021, doi:10.3390/molecules26030551_

Round 1

Reviewer 1 Report

Dear authors

lines 24-26, 39-43, 76-78 should be rewritten (the meaning is not clear). 

line 55, the word herein, or the phrase in this context should be erased.

line 92 reference is missing.

line 96 its effectively.

figures 2,7, are in low quality. The authors should provide better images.

A paragraph in the introduction is mandatory in order to answer the question why these two systems are important to be evaluated in such manner.

Best regards

Reviewer 2 Report

The paper presents a computational study on base metal complexes, and also verified their findings using simple measurements. While the study is interesting, the authors failed to illustrate the need for this study. So in the current form the study is more like a textbook exercise and do not warrant publication. I will try to categorize my comments below which may help the author to incorporate major changes prior re-submission.

I will illustrate the key points below:

  1. Please make a table with data from experimental and simulation so that we can compare the results (for charge distribution, electron density etc.). It’s hard to follow in current format.
  2. How well do fingerprint plots resemble the practical scenario?
  3. Authors have used multiple approximations to real scenario. A more comprehensive results need to be provided for how much % we are off from the true model
  4. Multiple grammatical errors are present in the paper and needs to be improved
